# NIRis: A low-cost, versatile imaging system for near-infrared fluorescence detection of phototrophic cell colonies used in research and education

**Ole Franz**, **Heikki Häkkänen, Salla Kovanen, Kati Heikkilä-Huhta, Riitta Nissinen, Janne A. Ihalainen**\*

Nanoscience Center, Department of Biological and Environmental Science, University of Jyväskylä, Jyväskylä, Finland

\* janne.ihalainen@jyu.fi

## Abstract

A variety of costly research-grade imaging devices are available for the detection of spectroscopic features. Here we present an affordable, open-source and versatile device, suitable for a range of applications. We provide the files to print the imaging chamber with commonly available 3D printers and instructions to assemble it with easily available hardware. The imager is suitable for rapid sample screening in research, as well as for educational purposes. We provide details and results for an already proven set-up which suits the needs of a research group and students interested in UV-induced near-infrared fluorescence detection of microbial colonies grown on Petri dishes. The fluorescence signal confirms the presence of bacteriochlorophyll *a* in aerobic anoxygenic phototrophic bacteria (AAPB). The imager allows for the rapid detection and subsequent isolation of AAPB colonies on Petri dishes with diverse environmental samples. To this date, 15 devices have been build and more than 7000 Petri dishes have been analyzed for AAPB, leading to over 1000 new AAPB isolates. Parts can be modified depending on needs and budget. The latest version with automated switches and double band pass filters costs around 350 € in materials and resolves bacterial colonies with diameters of 0.5 mm and larger. The low cost and modular build allow for the integration in high school classes to educate students on light properties, fluorescence and microbiology. Computer-aided design of 3D-printed parts and programming of the employed Raspberry Pi computer could be incorporated in computer sciences classes. Students have been also inspired to do agar art with microbes. The device is currently used in seven different high schools in Finland. Additionally, a science education network of Finnish universities has incorporated it in its program for high school students. Video guides have been produced to facilitate easy operation and accessibility of the device.

**Data Availability Statement:** All relevant data are within the paper and its Supporting information files.

**Funding:** RN and JI received a joined funding from Kone foundation under the project name "Shared light". Funding number is 202009816. The foundation website is https://koneensaatio.fi/en/. The funders did not and will not have a role in study design, data collection and analysis, decision to publish, or preparation of the manuscript.

**Competing interests:** The authors have declared that no competing interests exist.

## Introduction

Answering of research questions often benefits from niche tools, which are consequently not mass-produced as there is no sufficiently-sized market. This results in suitable research-grade devices being expensive and targeted for a broader range of applications. Here we introduce an affordable and open-source imaging system called NIRis (Near-infrared imaging system) which enables rapid screening of Petri dishes for phototrophic bacterial colonies. Building own equipment for the specific needs of research groups has become significantly easier with decreasing prices for electronic hardware and the recent development and increasing number of available and affordable 3D printers [1, 2]. Consequently, online resources for open-source lab equipment have increased steadily. Examples include various devices for microfluidics, imaging set-ups, plate readers, microscopes, as well as regular labware such as filter holders or pipettes [3–7]. Since its publication in 2012, Raspberry Pi single-board computers have been often integrated into these devices, as they are comparatively affordable and serve the needs for a broad range of applications [8]. Also in NIRis, the image acquisition and camera operation is performed by a Raspberry Pi computer coupled with a near-infrared (NIR)-sensitive camera module distributed by the same company. The various possibilities and advantages of Raspberry Pi computers have been previously described in detail and will not be further discussed in this article [8, 9]. Commercially available devices which could analyze samples to a similar level are advanced gel imaging chambers which are advertised to analyze Petri dishes as well. However, these machines are much larger, the customizability concerning emission and excitation light is limited and the price usually starts upwards of 10.000€. NIRis was build out of a need for a simple and efficient fluorescence detection chamber to reliably detect phototrophic bacterial colonies based on the presence of NIR-fluorescent bacteriochlorophyll $a$ (BChl $a$) [10–12]. The system records a reference white light image to capture all colonies, as well as a fluorescence image which selectively shows only phototrophic bacteria (Fig 1) Rapid and low-cost fluorescence detection devices are currently not commercially available, even

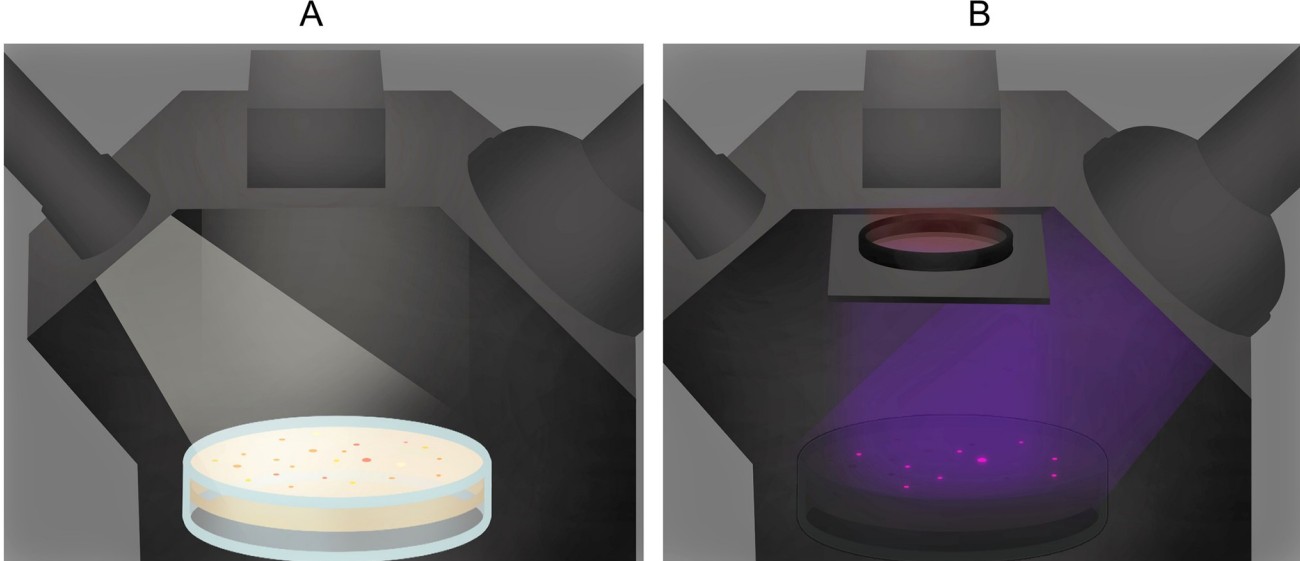

**Fig 1. Graphical illustration of NIRis in operation.** A: A white light reference image is captured to show the location of all colonies on the Petri dish. B: A UV-induced NIR fluorescence image is captured. Only NIR fluorescence coming from AAPB bacteria can pass the engaged filter and reach the camera on top for selective detection.

though the components required to build a suitable device for macroscopic analysis are relatively affordable. More advanced systems, like hyperspectral cameras or spectrometers can characterize fluorescence by providing detailed spectra, showing peak shapes and locations [13]. However, this information is not required if the primary goal is to identify and separate objects with a distinct difference in fluorescence for further analysis and more detailed characterization. Similar to fluorescence microscopy, it is possible to equip the imager with light sources of specific wavelengths or bandpass filters for targeted excitation and bandpass or longpass emission filters to analyze the spectral properties of cells for their classification. The obvious difference between a fluorescence microscope and NIRis is the sample size and resolution. This implies that microorganism need to be cultivated and grown into macroscopic colonies in order to be captured by the employed camera in NIRis. The advantage of the macroscopic approach is that selected colonies can be easily harvested from a Petri dish for further analysis. NIRis is therefore an extremely powerful tool in weeding through millions of colonies for those with specific spectroscopic features. In addition to the detection of naturally occurring fluorescence stemming from pigments, there may be also applications for the detection of fluorescent labels. For example, NIR-fluorescent probes were used to differentiate bacterial infections from tumors in mice [14]. In the following publication we provide details and results for an already proven set-up which suits both the needs of a research group and educational institutions interested in UV-induced near-infrared fluorescence detection of microbial colonies grown on Petri dishes.

## Materials and methods

### Assembly of the imager for UV-induced NIR fluorescence detection

The imaging system can be employed for a wide range of applications due to its modularity. The NIRis version detects the expression of light harvesting complex 1 (LH1) with integrated BChl *a* in the central reaction center of AAPB [11]. NIRis was inspired by a research-grade fluorescence imaging system used by Zeng and colleagues for the same purpose of identifying AAPB [15]. The major difference and advantage of NIRis is the use of low-cost consumer-grade hardware. The fixed sample, light and camera positions ensure repeatable imaging to allow for a more precise analysis of the images and a reduction of reflection problems. Additionally, this design provides portability without altering the imaging result.

The imager consists of a 3D-printed chamber holding each one consumer-grade white light-emitting diode (LED) and UV LED flashlight, each at an angle of 45 degrees to the sample. The chamber is ideally printed of black material and can be optionally painted matte black to reduce internal reflections and prevent light leakages. A simple diffuser is permanently placed in front of the white light to improve the illumination and reduce strong reflections from the Petri dish. The diffuser was cut out of readily available surgical masks made of polypropylene fibers. A bottom drawer can be removed from the chamber to insert a Petri dish sample for imaging (Fig 2). The employed 8-megapixel camera module PiNoir V2 (Raspberry Pi, Cambridge, UK) ships without a hot mirror allowing it to record also the NIR region of incoming light. It sits on top of the chamber and is operated by a Raspberry Pi computer model 3 B+ (Cambridge, UK) sitting on the back and a custom python software which enables camera acquisition time and gain changes to account for different light sources (S1 File).

The 19 parts needed to assemble the device were 3D-printed with glycol-modified polyethylene terephthalate (PETG) since it is known to be stronger than commonly used poly-lactic acid (PLA) filament and still easy to print (Fig 3). Parts were printed with a Prusa i3 Mk2 (Prague, Czech Republic) with a layer height set to 0.35 mm and 20% infill. All parts were assembled with M2 and M4 screws on a wooden board as a solid base. Computer-aided design

A                                                           B

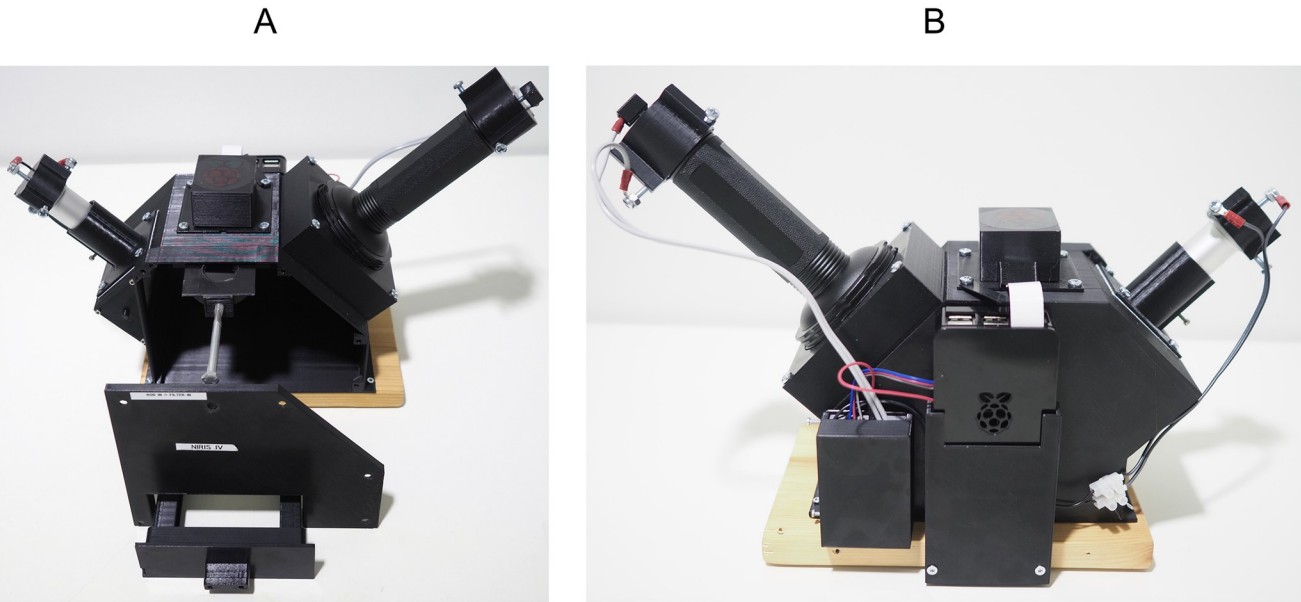

**Fig 2. Photos of the imager.** A: Front view of the 3D-printed imaging system with open front plate to show the sliding filter holder in the center, attached to the silver bolt. The white light source is mounted on the left and the UV excitation light source placed on the right. The sample Petri dish fits to the drawer at the bottom. B: Back of the imager showing the Raspberry Pi computer connection to the camera on top as well as the relay box on the left which operates the flashlights.

A                                                           B

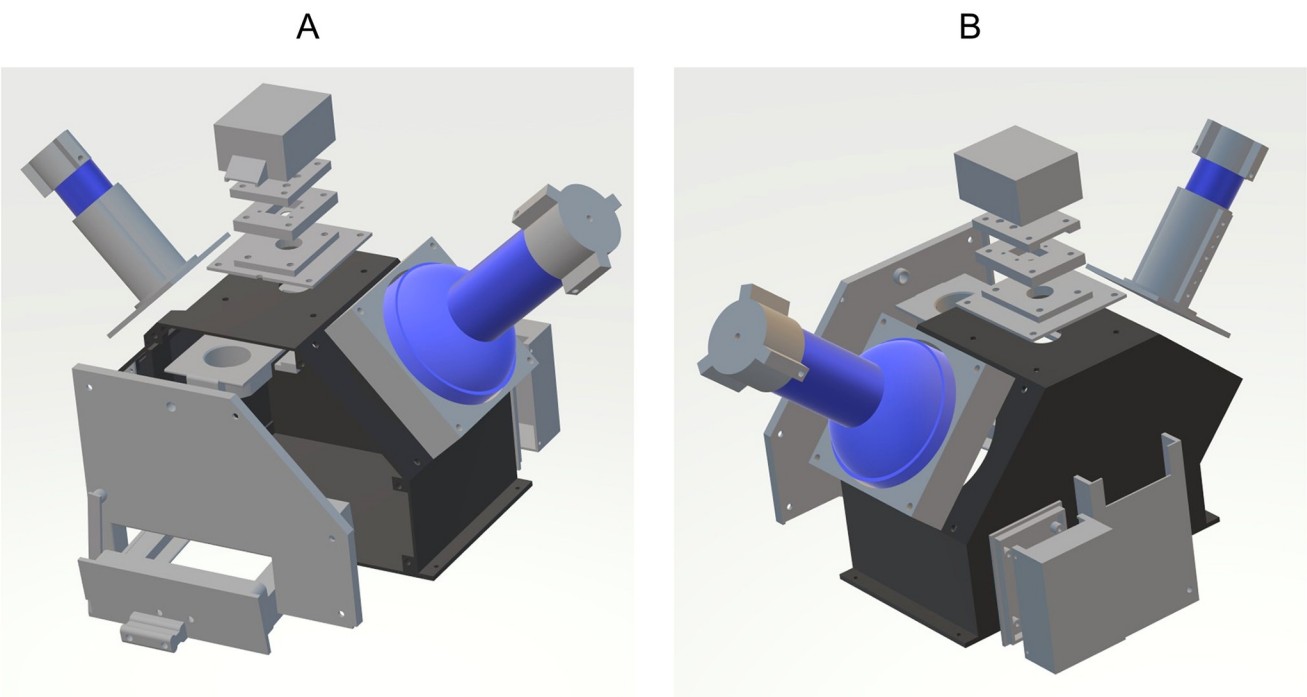

**Fig 3. Overview of 3D-printed parts.** All 3D-printed parts in grey, with the main chamber in dark and all other parts in light grey. Flashlight placement is indicated in blue. Other hardware components were not added. A: Exploded view from the top front. B: Exploded view from the top back.

(CAD) files for the individual 3D-printed parts, as well as a short description of each part can be found in the supplementary material (S1 Fig and S2 File).

The lighting has undergone several changes to improve its performance. Initially, the UV-excitation light was a small 12 LED flashlight with similar dimensions to the reference white LED flashlight. Along the way, improvements were done concerning the diffusion of the light sources, leading to a more even illumination which covers the entire Petri dish (S2 Fig). Also, in the first versions of the imager, both lights were controlled manually and a single excitation filter was employed. These versions were about three times as slow to operate compared to the latest iteration due to the manual operation of switches and the need for longer shutter speeds caused by the low power excitation light. Additionally, the clear identification of AAPB colonies was significantly harder, as more reflections bled through (S2 Fig). This meant that any automated counting and analyzing with imaging software was unreliable. The current set-up (September 2023) employs a 128 LED UV flashlight (Glossday, China) rated at 395 nm, powered by 6 AA batteries for the fluorescence excitation and a smaller, diffused white LED flashlight for the reference imaging mode. Both lights are still consumer-grade lamps to keep the price low and the acquisition easy. In the latest version of the imager, the flashlight switches are controlled through a python program, enabling the synchronization of light and image capture (Fig 1B). The switches are connected with the Raspberry Pi via a relay board (Fig 2). This optimization saved battery power and operation time drastically when imaging large amounts of plates. For detection of AAPB colonies, UV-induced NIR fluorescence can be employed which verifies the presence of BChl $a$ in the cells because it fluoresces at 890 nm, in-vivo [10] S3 Fig. For UV-induced NIR fluorescence imaging, a 880 nm ± 8 nm bandpass filter (Model FB880–70 with FWHM = 70 ± 8 nm, Thorlabs, Newton, NJ, USA) can be engaged in front of the camera restricting it to record only the expected fluorescence in the NIR region, from 850 nm to 920 nm (Fig 1, S3 Fig). This filter is placed inside the filter holder and can be operated with a bolt sticking out of the chamber side. Two filters can be stacked to further eliminate false positives from reflection bleed-through as their filtering characteristics multiply (e.g. 0.01% * 0.01% Transmission in blocking region). The hardware needed to build this version of the imaging system with two stacked filters and automated switches cost around 350€. The following table specifies the details and prices for the significant individual hardware components (Table 1).

The imager can also record a reference white light image to count all colonies and assess their size, shape and color. Additionally, it is easier to locate fluorescing colonies with a complimentary white light picture for subsequent isolation. For the reference image, the white light is turned on, the filter is pulled away from the camera and a picture is taken (Fig 1). Between the acquisition of both images, the Petri dish stays in the same position and both

**Table 1. Details of incorporated hardware parts.**

| Part | Brand | Part Number | Quantity | Date | Price |
|---|---|---|---|---|---|
| Rasp. Pi 3 B+ Premium Kit | Raspberry Pi | RS Pi 3B+ Kit | 1 | 06/2021 | 61.27€ |
| PiNoir Camera V2 | Raspberry Pi | 913–2673 | 1 | 06/2021 | 25.52€ |
| 128 LED UV Flashlight | Glossday | UV-128LED-FBA | 1 | 01/2022 | 20.45€ |
| 1" Bandpass Filter | Thorlabs | FB880–70 3 | 1 or 2 | 08/2021 | 99.34€ |
| 2-channel relay module | BerryBase | HLRELM-2 | 1 | 08/2021 | 3.60€ |

Prices were paid per unit on the indicated purchase date. Different white light LED flashlights were bought locally for around 5€ per piece. Only requirements for the flashlight were a diameter of 2.5 cm and a rear button switch to be able to connect the Raspberry Pi computer in case automated switches should be installed. Additionally, PETG filament for 3D printing and screws are needed for assembly.

flashlights remain at the same incident angle for every Petri dish which reduces the amount of variation caused by changing reflections and uneven lighting S1 Video. Additionally, a MATLAB script, which can be found in the supplementary material, can be run to superimpose both images to highlight the fluorescing colonies aiding isolation of the correct colonies (S3 File). Loading of the sample and the entire imaging process takes about 20 seconds per Petri dish. With the camera model, light sources and typical AAPB samples used here, acquisition times of 0.35 seconds at iso 800 are sufficient for adequate detection of phototrophic bacterial colonies. The python script which contains the imaging settings and handles the acquisition as well as the appropriate light switches can be also found in the appendix (S1 File).

## Sample preparation from plant endo- and phyllosphere

Detailed extraction and cultivation methods as well as isolation results have been published previously [11]. In short, phyllosphere bacteria were extracted by 3-minute sonication of sterile-collected plant material in potassium phosphate buffer with surfactant. Endosphere bacteria were extracted subsequently by sterilizing the plant surface with 3% sodium hypochlorite, triple water washing and maceration of the tissue in 20 mM KPi pH 6,5 buffer. Phyllosphere and endosphere extracts were plated in a dilution series and grown on half-strength R2A medium with additional 0.7% BD Bacto™ agar (Thermo Fisher Scientific, Waltham, MA, USA) adjusted to pH 6.5 with HCl, on 92x16 mm polystyrene Petri dishes (Sarstedt, Nümbrecht, Germany). Cells were grown for three days at room temperature after which they were placed in the fridge at +4˚C. Fluorescent signals of bacterial colonies could be detected as soon as three days after plating. However, some colonies needed several weeks to gain sufficient fluorescence for detection as the production of light harvesting complexes may require nutrient deficiency, light and temperature-related induction or other unknown factors.

## Results and discussion

### Identification of aerobic anoxygenic phototrophic bacteria with the imager

NIRis was primarily used to identify AAPB from environmental samples taken from plant phyllosphere and endosphere. As of now, over 7000 Petri dishes with samples from five countries and more than 20 locations have been imaged, resulting in over 1000 AAPB isolates. Operators ranged from high school interns, over B.Sc. and M.Sc. students to professionals indicating its ease of use. UV-induced NIR fluorescence can be employed for the identification of AAPB because it verifies the presence of BChl *a* which fluoresces at 890 nm, in-vivo [10]. For reference, a UV-induced fluorescence spectrum of the AAPB *S. glacialis* strain S2U11 can be found in the supplementary material (S3 Fig). Additionally, UV-induced fluorescence spectra of a variety of AABP have been published recently [11]. The fluorescence imaging successfully detected very small AAPB colonies reliably, at least from around 0.5 mm diameter and larger when measured using Fiji image software [16] (Fig 4, S4 Fig). The reference white light image taken with NIRis displayed all present colonies for their assessment of size and colour as well as enabling a total colony count (Fig 4). Additionally, it helped to locate fluorescing colonies with a complementary white light picture for subsequent isolation. White labels with printed sample codes can be helpful for sample identification as they are still readable in the fluorescent image, preventing accidental mix-ups (Fig 4). Fluorescence and white light reference images can be superimposed or viewed side by side in order to correctly identify fluorescing colonies. A MATLAB script used to superimpose the white light reference and fluorescence images highlighted AAPB colonies successfully (Fig 4, S2 File). Colonies of interest can then be isolated and re-streaked for fluorescence verification by NIRis or spectral analysis with more sophisticated and time-consuming methods (S3 and S4 Figs). The here

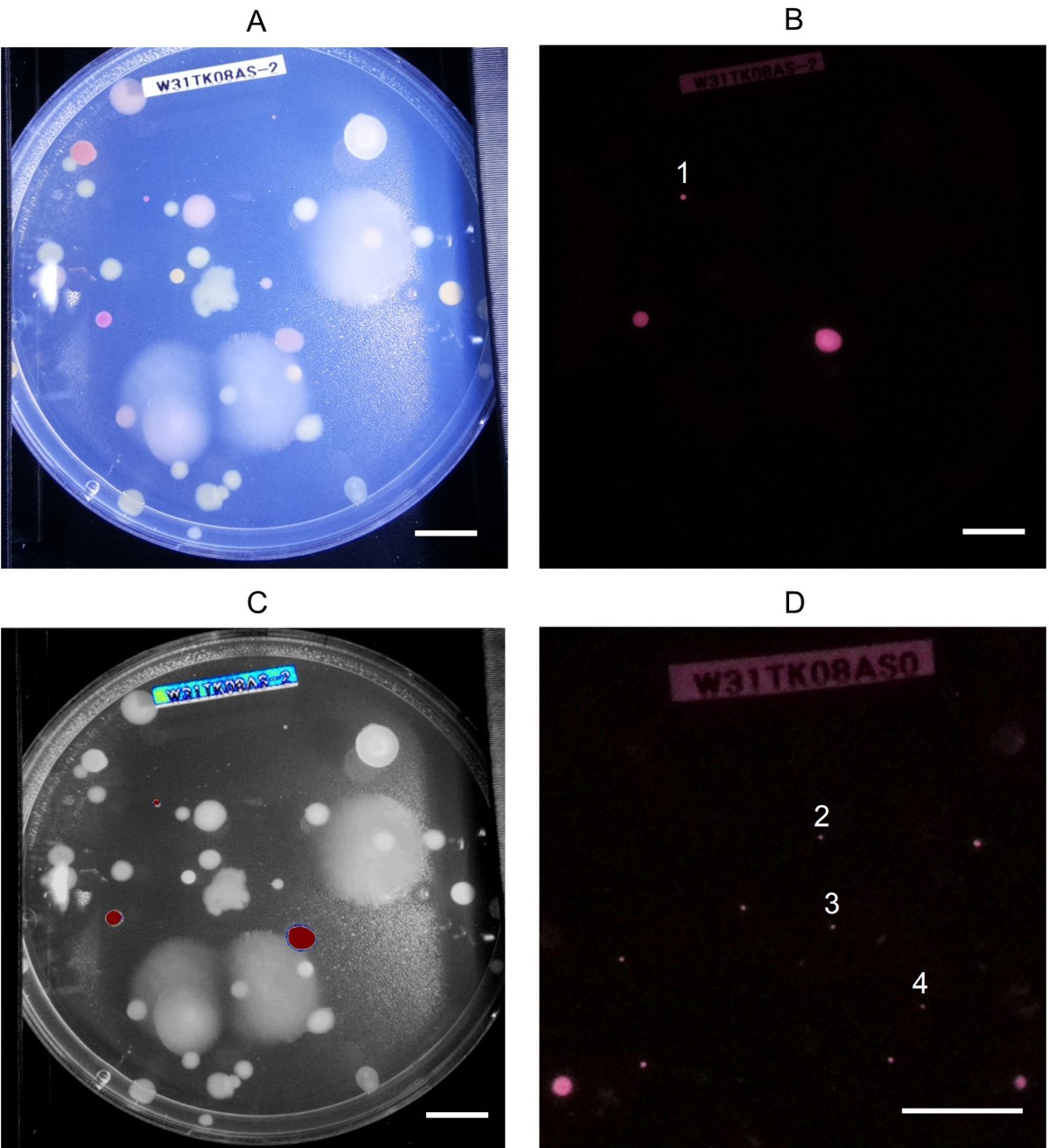

**Fig 4. Detection of AAPB colonies.** A: White light reference image of a typical Petri dish with a sample from plant phyllosphere taken with NIRis. B: The same plate in the same position as in A but imaged with engaged bandpass filter and UV flashlight. Diameter of colony underneath label 1: 0.99 mm C: Output of the MATLAB script which superimposed the images from A and B and highlights fluorescing areas to help pick the correct isolates. Blue = weak fluorescence, Red = strong fluorescence D: A magnified area of a different Petri dish with smaller AAPB colonies compared to A-C to demonstrate the resolution of NIRis in fluorescence mode. Diameters of colonies underneath labels 2, 3 and 4: 0.44 mm, 0.47 mm and 0.42 mm, respectively. Scale bars = 1 cm.

introduced workflow included UV-induced fluorescence spectra reading from the cell culture, collection of cells to frozen glycerol stock, as well as a separate material collection for DNA sequencing. The only non-AAP microbes which have been detected as positive with this imager so far have been microalgae. While chlorophylls in microalgae cause UV-induced

fluorescence peaks which are regularly under 700 nm, the tail of a strong fluorescence peak can bleed through the engaged bandpass filter into the 850–920 nm region. However, microalgae are usually rare in the here described samples and easy to tell apart from AAP bacteria, at the latest when the full spectra are recorded. During previous work, AAPB colonies identified with NIRis showed an absorbance peak at 872 nm, corresponding to the presence of BChl a [12]. Additional analysis of the isolates is currently under progress. Until now, all isolates which were picked and sequenced based on NIRis imaging harboured genes for the photosynthetic reaction center and showed a UV-induced fluorescence peak corresponding to BChl a in photosynthetic complexes [11].

## The imager in education

Cost-effective imaging devices that could be used for educational purposes are scarce—in particular devices, that could be constructed by high-school students themselves. Our project demonstrates the power of 3D-printing possibilities, relatively easy programming of the web camera, the properties of wavelengths and filters for imaging purposes and the nature of fluorescence. Also, the cultivation of environmental microbes on Petri dishes can be implemented in biology lectures to teach about the ubiquity, necessity and characteristics of bacteria in the surroundings and mitigate a fear of microbes [17]. CAD of 3D-printed parts and programming of the incorporated Raspberry Pi computer could be incorporated in computer sciences classes. These ideas have started to be realized in practice. As of winter 2023/24, imaging systems were supplied to 7 different high schools in Finland which participate in the project. Additionally, NIRis was incorporated to the program of a science education network of Finnish universities for high school students.

Students of all seven schools also participated in the seasonal sample collections across Finland, which was an important contribution to studying the ubiquity of AAPB in the phyllo- and endosphere of plants in boreal and subarctic environments [11]. The structure and content of upper secondary school studies in Finland is guided by the national core curricula and qualification requirements framework published by the Finnish National Agency for Education (EDUFI) [18]. According to a questionnaire answered by the participating teachers, four project schools in Jyväskylä, Espoo, Turku and Oulu have used NIRis during teaching in several courses in physics, biology and visual arts (Table 2). The courses employing NIRis as named by the teachers, as well as their references in the EDUFI publication are listed in Table 2.

Table 2. Courses using NIRis named by participating teachers.

| Course Code | Course name | Reference |
|---|---|---|
| Physics FY7 | Electromagnetism and light | EDUFI 2019, p. 256 |
| Physics FY8 | Matter, radiation and quantization | EDUFI 2019, p. 257 |
| Biology BI1* | Life and evolution | EDUFI 2019, p. 236 |
| Biology BI2* | Basics of ecology | EDUFI 2019, p. 237 |
| Biology BI4 | Cell and genetics | EDUFI 2019, p. 238 |
| Biology BI6 | Biotechnology and its applications | EDUFI 2019, p. 240 |
| Visual arts KU4 | The many worlds of art | EDUFI 2019, p. 350 |

All national courses that teachers from participating schools in Jyväskylä, Oulu, Espoo and Turku have used NIRis in so far. EDUFI refers to the publication by the Finnish National Agency for Education from 2019 concerning the study plans for upper secondary schools in Finland [18].

* = compulsory courses for all students

NIRis has been most extensively employed for teaching by the Jyväskylä Lyseo upper secondary school, which generally emphasizes teaching natural sciences and works in close collaboration with the University of Jyväskylä. So far, NIRis was used especially in the school-specific physics practical courses and in the astronomy course. In addition, the physics and biology teachers have planned and implemented a school-specific interdisciplinary study course based on the NIRis project involving plant-inhabiting AAPB and the electromagnetic spectrum of light. This course is held annually in August as a compact weekend seminar at the University of Jyväskylä's research station in Konnevesi. Students who complete this course can receive credits in their possible future university studies, for example when starting to study in the Nanoscience degree at the University of Jyväskylä. In the years 2021–2023 about 120 students from Jyväskylä Lyseo upper secondary school have participated in courses in which the NIRis was included.

In the upper secondary school in Oulu, NIRis has also been used in the national compulsory biology courses BI1 (Life and evolution) and BI2 (Basics of ecology), so that all students get to know the subject (Table 2 and [18]). When collecting samples for the researchers of the project, the students have also collected their own samples, which have been grown on Petri dishes and imaged with NIRis to map AAPB. Thus, it has been possible to carry out independent research projects on the presence of AAPB. The BI6 course aims to plan and implement experimental work related to the application of biology [18]. Also in Turku, the students of the BI6 course have realized a small-scale research project in which the presence of AAPB in different plant species has been mapped. For that, the imager was used to analyze bacteria which grew after plant leaves or branches were pressed on nutrient agar in Petri dishes. This technique was first described by William Corpe in 1985 and has been already successfully implemented in teaching environments before [17, 19]. Surface bacteria on leaves often include AAPB which can be harvested by the leaf printing method and identified as AAPB by UV-induced NIR fluorescence imaging using NIRis (Fig 4). The project work included sterile sampling, working with Petri dishes and bacterial colonies as well as an introduction to the research question. Eight comprehensive video guides aimed for the teachers and students, along with a shorter video demonstration for this publication have been produced to explain all parts of the process (S1 Video and [20]). The students have indeed discovered previously undocumented woody species containing AAPB. That data is currently under preparation and will be published in the near future.

Additionally, students have been inspired to do agar art with microbes which could be later imaged with the device. For creating agar art, differently colored bacteria strains are used as paints to create images on Petri dishes. There is even more room for creativity, if selected strains are fluorescent which selectively enhances their appearance under excitation light (Fig 5). Here, these strains are AAPB, which will be visible in NIRis imaging when using the fluorescence mode. The topic has been popular—as shown in the yearly agar art competition held by the American Society for Microbiology [21]. Additionally, agar art has been discussed as an educational tool for undergraduate students, for example to teach about fluorescent protein expression [21, 22]. Agar art and subsequent imaging with NIRis has also been part of the European Researchers´ Night event at the University of Jyväskylä. Participants could paint a picture on an own Petri dish or leave a signature on an agar guest book. After one week of incubation time, images were taken both in white light and fluorescent mode and sent to the participants. It can be noted that the white light image produced by NIRis is not suitable if an appealing picture with vibrant colours, good contrast and lighting is required, as it would be for example in the presentation of agar art. Therefore a pocket camera or digital single-lens reflex (DSLR), paired with suitable diffused lighting can be used additionally to enhance the appearance (Fig 5). However, the low cost camera integrated in NIRis is sufficient for its

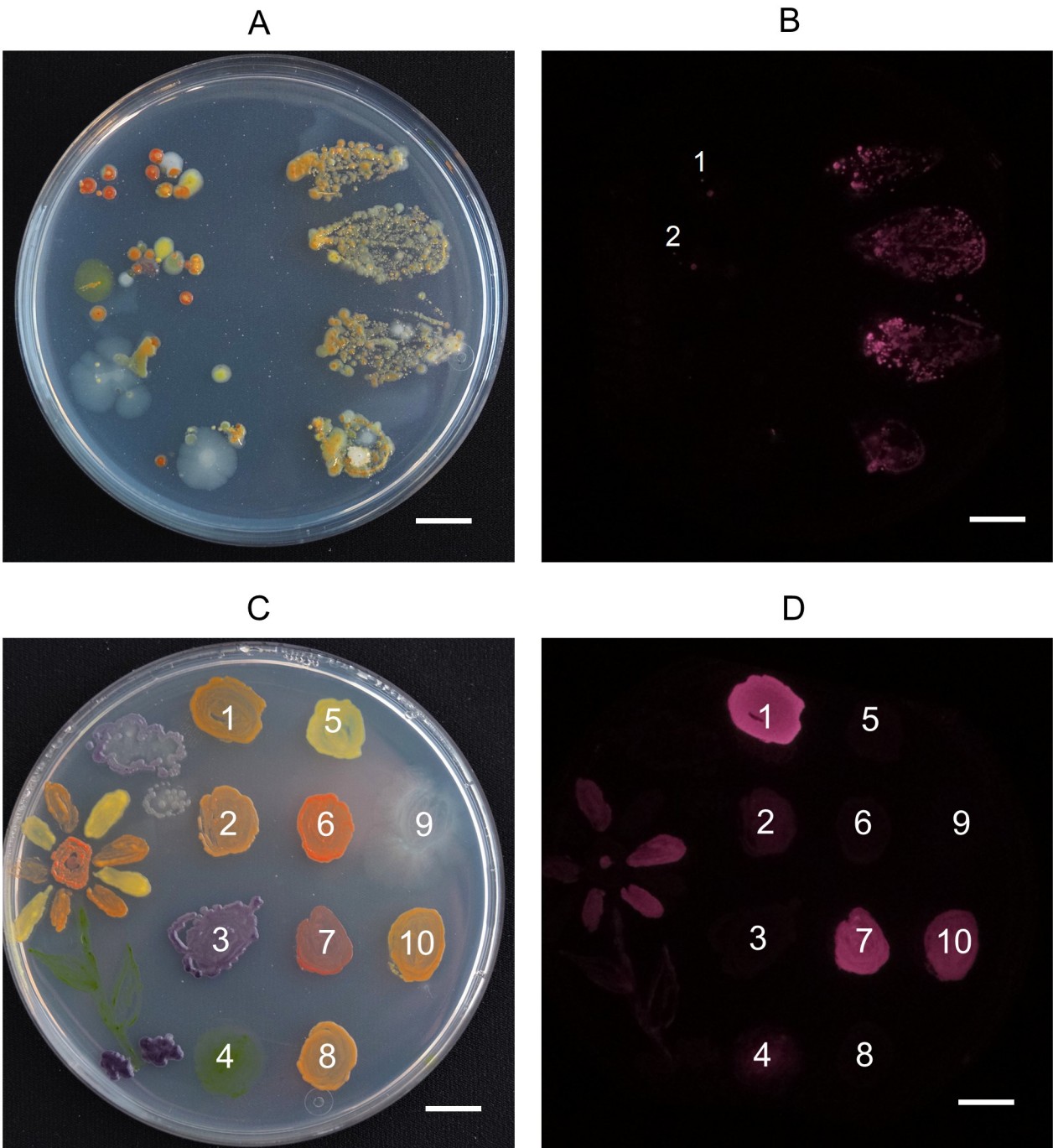

**Fig 5. The imager in education.** A: Four lingonberry (*Vaccinium vitis-idaea*) leafs were placed first top down on the left side and then flipped over with the underside down on the right side of the agar plate. The image was taken after incubation for 4 days at room temperature and subsequently 9 days at +4°C with a DSLR for better clarity. B: Fluorescence image of the same Petri dish taken with NIRis right after the image A was taken. Violet areas show the location of AAPB colonies. Colonies underneath label 1 and 2 were measured to have diameters of 0.44 mm and 0.46 mm, respectively. C: Colour palette for agar art photographed with a DSLR for better clarity and colour reproduction. Species specification: 1 *Sphingomonas faeni* 2 *Aureimonas* sp. 3 *Collimonas* sp. 4 *Stichococcus* sp. (microalgae) 5 *Sphingomonas faeni* 6 *Rhodococcus* sp. 7 *Methylobacterium* sp. 8 *Sphingomonas faeni* 9 *Pseudomonas antarctica* 10 *Sphingomonas glacialis* D: Same Petri dish as shown in C but imaged in fluorescence mode with NIRis. Numbers correspond to the same species specified in C. Scale bars = 1 cm.

primary purpose—to be superimposed with the fluorescent image for the localization of fluorescent colonies. In addition to presenting the various colors of microorganisms, the colour palette also demonstrates the difference of pigmentation in the same species *Sphingomonas faeni*. Patch numbers 1, 5 and 8 have been sequenced to be the same species but show a clear difference in color and UV-induced NIR fluorescence (Fig 5). The picture also demonstrates that the fluorescence of chlorophyll in micro algae can show in the UV-induced NIR fluorescence image as discussed before (Fig 5 patch 4).

## Overview and comparison of open-source imaging systems

Several impressive open-source imaging systems have been published in the recent years. Nuñez and colleagues have developed an open-source device which could likely identify AAPB when modified to record UV-induced NIR fluorescence [23]. At the time of designing NIRis, we were unaware of existing open-source imaging devices. The design was inspired by a sketch of an imaging set-up in the appendix of a publication by Zeng and colleagues in 2014 [15]. They utilized a research-grade CCD sensor with a similar light and filter arrangement to detect AAPB on Petri dishes. However, they employed LED light in the blue-green spectral region (450 570 nm) with the idea to utilize the effect of energy transfer from carotenoids to BChl *a* to induce NIR fluorescence [15]. In NIRis the excitation light is around 400 nm where the so-called Soret bands of BChl absorb. Through internal conversion, BChl returns to the first electronic singlet state (S1). Transition to the ground state result in fluorescence emission in the NIR region. This mechanism consequently does not rely on carotenoid energy transfer and should more directly detect BChl *a*. This could be important as the fluorescence excitation spectrum of the AAPB *S. glacialis* strain S2U11 indicated that such energy transfer was hardly observable [12]. The difference between the already published open-source imager by Nuñez and colleagues is the light source placement and grade of complexity. While Nuñez and colleagues, as well as Gonzales and colleagues with their agarose gel imager, opted for a trans-illuminance LED circuit board underneath the Petri dish, the here presented imager utilizes flashlights for white light epi-illumination and UV-light excitation [6, 23]. The imaging system employs a simple push-in emission filter to switch between white-light reference and fluorescence imaging modes.

An arguably more complex open-source spectral imaging device was published by Lien and colleagues including the possibility for automated hyperspectral imaging of macroscopic samples [24]. However, the use of a hyperspectral camera places it in a different price category. Likewise more sophisticated is an open-source fluorescence imaging device which utilizes filter cubes, an incubation chamber and a consumer-grade DSLR camera with macro lens [25] Besides the higher costs, this set-up is restricted to the detection of signals in the visible light range, unless the internal hot mirror of the camera is removed.

## Customizations and modularity

The here provided instructions and CAD files enable anyone with access to a 3D printer to recreate the introduced imaging system. However, we hope that the presented approach to design custom imaging solutions would inspire others to develop tailored solutions for their specific needs. 3D-printing is a versatile method enabling low-cost custom solutions suitable for research groups as shown by a range of publications [3–7] The system can and has been optimized and simplified for different applications. If necessary, flashlights as light sources combined with pocket cameras can enable wireless operation to ensure complete portability for field use. While pocket cameras would provide the necessary resolution and light sensitivity for the detection of fluorescent colonies, they are generally equipped with hot mirrors in front

of their sensors to filter out NIR and infrared light. This mirror can be manually removed but it is usually more difficult than using the Raspberry Pi solution presented here. However, we encourage research groups which have a need for a recorded emission wavelength in the visible light spectrum to try and utilize regular consumer cameras for an even easier and more portable approach. The here presented imaging system could be assembled even more economically if acrylic NIR-transmitting filters are used instead of research quality filters, as these are the most expensive part. Those filters are easily available as they have been used extensively for shading of remote-control receivers, e.g. in televisions. Also, many 3D-printed parts are not vital for the correct operation of the system and can be omitted for more affordable and faster construction. For example, circuit board and camera module covers have been only added to guarantee resilience to withstand school use and easy transportation, as well as a neater appearance. Parts that were deemed non-vital for normal operation were marked as such in the parts list (S1 Fig). Of course, all hardware components can be exchanged by re-modelling the fixtures on the chamber. For example, sample holders for 96 well plates or holders for different flashlight sizes are possible to be designed using CAD software. An optional holder for an excitation filter could be attached in front of the excitation light source. Also the connected, automated flashlight switches are not necessary and can be omitted for simplicity, similar to how older versions of the imager operated. For the most stable light output a wired power supply for the UV flashlight can be built to minimize power fluctuations caused by battery operation and heating of LEDs. This has also been successfully tested. However, batteries have lasted weeks in our use after the electronic switches were installed, so the current standard version of the imager still works with battery-powered flashlights.

## Conclusions

Our research group is investigating potential habitats of aerobic anoxygenic phototrophic bacteria (AAPBs). The successful development of the imaging device enabled us to do a large-scale screening of bacteria harbouring photosynthetic complexes. BChl *a* present in these bacteria fluoresces in the NIR region and can consequently act as a natural marker for AAPB. The imager was used to analyze over 7000 samples from five different countries with an emphasis on samples from Finland with the first results being published recently [11]. Plant samples were collected from their natural habitats and bacteria from the phyllosphere and endosphere of leaves and twigs could be cultivated. NIR fluorescent bacteria were efficiently detected by the imaging device and could be isolated to create pure cultures for sequencing and further analysis [11]. With our settings, recording of a fluorescence and white reference image of a single Petri dish took about 20 seconds and enabled high throughput screening. The low cost of the device enabled the integration of citizen science to our project. So far seven different high schools in Finland have received an imaging device to be able to analyze their own samples for AABP bacteria. Students could learn about light properties, photosynthesis, fluorescence and sterile microbiology work flows. The device can also easily spark interest in 3D printing, coding and of course in a career in natural sciences. Hands-on learning has been proven to be a successful strategy and can be a welcoming change to theoretical classes. Additionally, we assume that the participation in a research project may enhance the engagement of the students. The main advantages of NIRis are the low cost, simplicity and NIR recording capabilities as well as its portability. While more complex devices may enable additional applications, this imager performs very well for the identification of fluorescent colonies on Petri dishes— its primary purpose. Open-source research instruments like this imaging system essentially reduce the costs of scientific work and therefor lower the barrier for research groups with limited funds to contribute and participate in scientific discourse. We believe that developing

easily accessible lab instruments and the subsequent publishing of instructions can contribute directly to more democratized and inclusive science.

## Supporting information

**S1 Fig. 3D-printed parts and short assembly instructions.** Parts are rotated to display their construction. Parts with numbers in red are optional. 1 & 8: Flashlight caps to access the switches. 2 & 7: Flashlight holders 3: Camera assembly cover 4–6: Camera assembly parts. Camera module is placed between part 5 & 6. 9 & 11: Front wall 10: Raspberry Pi holder to be mounted to the back. 12: Main chamber 13: Security clamp (2X), can be mounted in two positions in front and on top to prevent Raspberry Pi and the sample drawer from sliding out. 14: Mount for filter holder—to be fixed underneath the top opening. 15: Filter holder, slides into mount (14). A bolt is secured in the front hole reaching through the corresponding hole of the front plate so that the filter holder can be engaged and disengaged 16: Cover for the relay board. 17: Base plate holding the relay board—to be fixed at the back. 18: Sample drawer for Petri dishes 19: Handle for the sample drawer.
(TIF)

**S2 Fig. Image examples taken with different iterations of NIRis.** A & B: White light and fluorescence images taken with an early iteration of NIRis, which had a weaker 12 LED, non-diffused excitation light and a single excitation filter. The reliable identification of fluorescent colonies across the entire plate is hard. Reliable automation of the AAPB identification would be impossible. C & D: White light and fluorescence images with NIRis which employed a sufficiently diffused and weaker 12 LED light source with a single excitation filter. Reflections of water droplets on the lid and negative colonies bleed through making it difficult to automate the identification with software. E & F: Current set-up of NIRis with a strong 128 LED excitation lamp and two stacked excitation filters. Even though the sample has large white colonies and water droplets on the lid, only truly fluorescent colonies are visible in the fluorescent mode. Scale bars = 1 cm.
(TIF)

**S3 Fig. Fluorescence spectrum of a typical AAPB.** An UV-induced fluorescence spectrum recorded from *Sphingomonas glacialis* strain S2U11 showing the approximate wavelength range passing through the 880 nm ± 8 nm bandpass filter (FWHM = 70 ± 8 nm) to the sensor (white area). The imaging set-up combined the same excitation flashlight rated at 395 nm used in NIRis with a NIR-sensitive iDus InGaAs Spectroscopy CCD Camera (Andor Technology, Belfast, Ireland) as described in [12].
(TIF)

**S4 Fig. NIRis images of isolated AAPB.** A: White light reference image of a strain of *Sphingomonas faeni* streaked out for fluorescence confirmation. B: The UV-induced NIR fluorescence image of the same Petri dish as in A, showing a strong fluorescence of the strain. C: A magnified white light reference image of three unidentified isolates, together with a regular ruler with millimeter graduation to emphasize the scale. D: The same Petri dish and frame as in C but imaged in the fluorescence mode. Both pink strains are NIR fluorescent, the white strain is not. The diameters of the smallest positive colonies are clearly below 0.5 mm. Scale bars = 1 cm.
(TIF)

**S1 File. Python script.** The file contains the python script which handles image acquisition as well as the flashlight switches. Image acquisition time and camera gain can be changed in the

script to account for different light sources or sample intensities.
(PY)

**S2 File. CAD files of NIRis parts.** The folder contains CAD files for all individual parts
needed to construct the imaging system. Files are numbered according to the numbering in
S1 Fig.
(ZIP)

**S3 File. MATLAB script.** The m-file contains a MATLAB script which superimposes two
images with the same name but different prefix; f (fluorescence) and g (reference). Fluorescent
areas are highlighted.
(M)

**S1 Video. Operation of NIRis.** A short demonstration video showing the basic operation of
NIRis.
(MP4)

## Acknowledgments

The participating high schools classes and their teachers from Utsjoen saamelaislukio, Rova-
niemen Lyseon lukio, Kuusamon lukio, Oulun steinerkoulun lukio, Jyväskylän lyseon lukio,
Otaniemen lukio and Turun Kerttulin lukio are appreciated for participating in the project
and giving feedback about the device. We thank Marleena Reponen for help with the elec-
tronic assembly.

## Author Contributions

**Conceptualization:** Heikki Häkkänen, Riitta Nissinen, Janne A. Ihalainen.

**Data curation:** Ole Franz, Janne A. Ihalainen.

**Formal analysis:** Janne A. Ihalainen.

**Funding acquisition:** Kati Heikkilä-Huhta, Riitta Nissinen, Janne A. Ihalainen.

**Investigation:** Riitta Nissinen, Janne A. Ihalainen.

**Methodology:** Ole Franz, Heikki Häkkänen, Kati Heikkilä-Huhta, Riitta Nissinen, Janne A.
Ihalainen.

**Project administration:** Kati Heikkilä-Huhta, Riitta Nissinen, Janne A. Ihalainen.

**Resources:** Riitta Nissinen, Janne A. Ihalainen.

**Software:** Heikki Häkkänen.

**Supervision:** Riitta Nissinen, Janne A. Ihalainen.

**Validation:** Ole Franz, Riitta Nissinen, Janne A. Ihalainen.

**Visualization:** Ole Franz, Salla Kovanen.

**Writing – original draft:** Ole Franz.

**Writing – review & editing:** Ole Franz, Janne A. Ihalainen.

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
