## [Decision Letter · Decision Letter 0]

5 Sep 2023

PONE-D-23-14942NIRis: A low-cost, versatile imaging system for NIR fluorescence detection of phototrophic cell colonies used in science and educationPLOS ONE

Dear Dr. Franz

Thank you for submitting your manuscript to PLOS ONE. After careful consideration, we feel that it has merit but does not fully meet PLOS ONE’s publication criteria as it currently stands. Therefore, we invite you to submit a revised version of the manuscript that addresses the points raised during the review process. 

We look forward to receiving your revised manuscript.

Kind regards,

Omnia Hamdy, PhD

Academic Editor

PLOS ONE

Journal Requirements:

Reviewers' comments:

Reviewer's Responses to Questions

**Comments to the Author**

1. Is the manuscript technically sound, and do the data support the conclusions?

Reviewer #1: No

Reviewer #2: Partly

2. Has the statistical analysis been performed appropriately and rigorously? 

Reviewer #1: No

Reviewer #2: No

3. Have the authors made all data underlying the findings in their manuscript fully available?

Reviewer #1: Yes

Reviewer #2: Yes

4. Is the manuscript presented in an intelligible fashion and written in standard English?

Reviewer #1: Yes

Reviewer #2: Yes

5. Review Comments to the Author

Reviewer #1: The paper proposes a low-cost equipment design for near infrared imaging applied to detection of phototrophic cell colonies. I believe it is a nice contribution for the scientific community, but I have some concerns about the presentation. I believe the paper is not ready for publication.

Specific comments:

- The paper contains the bill of materials well described and the promising resulting imaging. However, it is not much scientific. I would expect to see in a paper like this how the equipment was designed, a computational mathematical model, some analysis, maybe some optimization. However, the paper looks like a datasheet of a product with instructions on how to build it.

- I suggest not using acronyms in the title, even well-known ones like NIR.

Reviewer #2: This manuscript describes the use of off-the-shelf components in conjunction with desktop 3D printing to manufacture a portable, and accessible tool for the evaluation and detection of UV-activated IR fluorescence. The manufactured device has been applied for the detection of bacteriochlorophyll a as a marker for the presence of AAPB colonies. The simplicity of the design and assembly has enabled the integration of such tool in different educational settings which qualify it as a resource for both educational and research activities. Although this may be interesting to wide spectrum of audience, this reviewer is not sure about the suitability of this work for PLOS one, this reviewer thinks this may fit better into one of the chemical education journals. In addition, there are few issues that need to be addressed before this article is published including:

• The conclusion that the device is capable of detecting colonies as small as 0.5 mm is not supported by the data presented. Ideally, proper analysis of the detection limits of the device should be included.

• While demonstrating the applicability of the device in different settings is a good approach to highlight it accessibility, none of the applications presented discuss robustness and accuracy of the device in collecting emitted fluorescence. This has to be addressed using, for example, one of the standard fluorescent dyes at different concentrations across few different devices.

• Authors need to comment on the need to switch to other imaging devices (for example DSLR in Fig 5) in order to get clearer images.

6. PLOS authors have the option to publish the peer review history of their article (what does this mean?). If published, this will include your full peer review and any attached files.

Reviewer #1: No

Reviewer #2: No

---

## [Author Response · Author response to Decision Letter 0]

27 Nov 2023

Reviewer 1

1.1 The paper contains the bill of materials well described and the promising resulting imaging. However, it is not much scientific. I would expect to see in a paper like this how the equipment was designed, a computational mathematical model, some analysis, maybe some optimization. However, the paper looks like a datasheet of a product with instructions on how to build it.

We agree with reviewer 1 that not much information was given on the design process. We added sections addressing some of the key points during development, as well as results from previous iterations of the device (e.g. Appendix, Fig. S2). Reviewer 1 also commented that the submitted manuscript is not much scientific. We think that this partly concerns the unusual structure of the manuscript, which does not strictly follow the most common approach of answering a research question with available equipment. Instead, we want to introduce new research equipment and make it available for the scientific community. Therefore, the emphasis was put on providing all necessary information to build that design, making it look like the “datasheet (…) with building instructions” it is. We have no computational mathematical model but believe that the results are nonetheless useful for the research community. We have seen similar publications in PLOS One and therefore decided to submit to this journal.

1.2 I suggest not using acronyms in the title, even well-known ones like NIR.

We agree with reviewer 1 and have changed the title accordingly.

Reviewer 2

2 (…) this reviewer thinks this may fit better into one of the chemical education journals.

It was hard to categorize this manuscript because of its multi-disciplinary nature. We have considered the possibility to submit to an educational journal. However, we have no significant qualitative data from the participating schools and want to stress that the imager is indeed vital for everyday operations in our research group. It may be that in the future, we will collect more data from participating schools and publish that data set separately, in collaboration with education researchers. We have noticed similar, multi-disciplinary articles spanning equipment development, research and education published in PLOS ONE and found it best suitable for our submission. 

2.1 The conclusion that the device is capable of detecting colonies as small as 0.5 mm is not supported by the data presented. Ideally, proper analysis of the detection limits of the device should be included.

Initially we believed that the scale bar in figure 4 would provide sufficient evidence that the imager records fluorescent colonies in that size range. The scale bar was done accurately with Image J using a known reference length (the length of the white label on the Petri dish). However, we now added an additional figure with more pictures taken with the device, including a set with an included ruler (Fig. S4).

The difficulty in a “proper analysis of the detection limit” is, that the detection limit is likely not only dependent on the size, but also on the number and brightness of fluorescent molecules in the cells, the total number of cells as well as cell density in the colony, all of which is very hard to analyze or predict. Therefore, artificial fluorescent samples with clear dimensions are unsuitable to assess the detection limit of fluorescent AAPB colonies. However, we hope that the new figure gives better insights into the capabilities of the device. 

2.2 While demonstrating the applicability of the device in different settings is a good approach to highlight it accessibility, none of the applications presented discuss robustness and accuracy of the device in collecting emitted fluorescence. This has to be addressed using, for example, one of the standard fluorescent dyes at different concentrations across few different devices.

We have considered a lot about how to make a comparison that makes sense. While we agree that it would be nice to show fluorophores with the imager, there are several difficulties regarding their uses. 

(1.) The amount of fluorescence emitted and captured by the camera does not only depend on fluorophore concentration, but also on the dimensions and properties of the imaged object. A high cone will result in a different fluorescence intensity than a flat droplet. Additionally, measuring suspended molecules will differ from measuring colonies. In our opinion there is no use in assessing the capabilities to detect fluorescent colonies by using samples of different nature.

(2.) To our knowledge there is no standard fluorophore which would be excited in the UV wavelength and emits in the NIR wavelength, similar to how AAPBs are detected by us. For example, using suspended Bacteriochlorophyll a as a test would be unsuitable as its fluorescence properties are different when extracted compared to being incorporated in the light harvesting complex inside a cell (fluorescence maximum at 780-800 nm in organic solution and 890-910 nm in LH1 complexes; employed excitation bandpass filter 880 nm ± 8 nm (FWHM = 70 ± 8 nm)). Changing light sources and excitation filters to match common fluorophores will unavoidably change the performance of the imager and make it unsuitable for an evaluation of its AAPB recording capabilities. Also, the camera sensors´ sensitivity will differ at different wavelength. 

Comparing the same sample across different devices proves to be difficult. Imaging devices and spectrometers can be tweaked in numerous ways making a comparison difficult. Increase in sensor gain (iso), acquisition time or excitation intensity will affect the result. There is no standard device or method that would do the same AAPB colony identification as NIRis, which is why we cannot compare it to a standard method.

Concerning the robustness of the device, we believe that the amount of Petri dishes that have been imaged so far, in combination with the amount of successfully isolated AAPBs, give a hint on how robust our system is. The referenced pre-print of a separate work from our group (Nissinen et al., 2023) provides a documented excerpt which provides more detailed analysis of the isolates identified with NIRis. Additionally, previous work demonstrated that all AAPB colonies detected with the device showed a BChl a absorbance peak when imaged with a hyperspectral camera (Franz, 2022). This information was added to the manuscript. 

2.3 Authors need to comment on the need to switch to other imaging devices (for example DSLR in Fig 5) in order to get clearer images.

The use of a DSLR simply provides cleaner, more vibrant colours and better detail than the 25€ raspberry pi camera. The white light images taken with NIRis are still necessary and suitable for the location of fluorescent signals on a sample plate. It is however not suitable for the aesthetic presentation of agar art, which benefits from a better and more expensive camera set-up, including larger diffused lighting than it would be possible in our device. We have added some remarks in the manuscript to clarify this matter.

---

## [Decision Letter · Decision Letter 1]

26 Dec 2023

PONE-D-23-14942R1NIRis: A low-cost, versatile imaging system for near-infrared fluorescence detection of phototrophic cell colonies used in research and educationPLOS ONE

Dear Dr. Franz,

Thank you for submitting your manuscript to PLOS ONE. After careful consideration, we feel that it has merit but does not fully meet PLOS ONE’s publication criteria as it currently stands. Therefore, we invite you to submit a revised version of the manuscript that addresses the points raised during the review process.

We look forward to receiving your revised manuscript.

Kind regards,

Omnia Hamdy, PhD

Academic Editor

PLOS ONE

Journal Requirements:

Reviewers' comments:

Reviewer's Responses to Questions

**Comments to the Author**

1. If the authors have adequately addressed your comments raised in a previous round of review and you feel that this manuscript is now acceptable for publication, you may indicate that here to bypass the “Comments to the Author” section, enter your conflict of interest statement in the “Confidential to Editor” section, and submit your "Accept" recommendation.

Reviewer #1: All comments have been addressed

Reviewer #2: (No Response)

2. Is the manuscript technically sound, and do the data support the conclusions?

Reviewer #1: Yes

Reviewer #2: Partly

3. Has the statistical analysis been performed appropriately and rigorously? 

Reviewer #1: N/A

Reviewer #2: N/A

4. Have the authors made all data underlying the findings in their manuscript fully available?

Reviewer #1: Yes

Reviewer #2: Yes

5. Is the manuscript presented in an intelligible fashion and written in standard English?

Reviewer #1: Yes

Reviewer #2: Yes

6. Review Comments to the Author

Reviewer #1: All my comments have been addressed. As far as I could verify, the paper meets the requirements of PLOS ONE to be published. It still looks like the datasheet of a product with instructions on how to build it, but I believe it may be useful for the scientific community.

Reviewer #2: Revised manuscript does address some of the previously raised comments, but the manuscript may need few changes before publication.

1- Can you add scale bars (or dimensions to all figures)

2- In order to proper assess the detection limit of the system, you may be able to present colonies/cells in a better way. For example, by defining the borders of each colony and adding approximate dimensions (particularly for figures 4 and 5)

3- While manuscript claims that the device has been in use in 7 different high schools in Finland, it did not include any supporting information on its applicability in schools. Ideally this may include some data form students-lead projects or at least describe how this has been integrated within the program/curriculum.

7. PLOS authors have the option to publish the peer review history of their article (what does this mean?). If published, this will include your full peer review and any attached files.

Reviewer #1: No

Reviewer #2: No

---

## [Author Response · Author response to Decision Letter 1]

6 Feb 2024

Reviewer #2: Revised manuscript does address some of the previously raised comments, but the manuscript may need few changes before publication.

1- Can you add scale bars (or dimensions to all figures)

Response: Scalebars have been now added individually to each of the images taken with the NIRis device. 

2- In order to proper assess the detection limit of the system, you may be able to present colonies/cells in a better way. For example, by defining the borders of each colony and adding approximate dimensions (particularly for figures 4 and 5) 

Response: We have now included measurements for selected colonies to provide more accurate data on the diameter of observable colonies. The measurements were done accurately with Fiji image software using a known distance to set the scale and subsequently measure diameters in each image individually.

3- While manuscript claims that the device has been in use in 7 different high schools in Finland, it did not include any supporting information on its applicability in schools. Ideally this may include some data form students-lead projects or at least describe how this has been integrated within the program/curriculum.

Response: We have now included extensive information on the use of NIRis in the curriculum of participating schools, based on the teacher´s information and experiences.

---

## [Decision Letter · Decision Letter 2]

1 May 2024

NIRis: A low-cost, versatile imaging system for near-infrared fluorescence detection of phototrophic cell colonies used in research and education

PONE-D-23-14942R2

Dear Dr. Franz,

We’re pleased to inform you that your manuscript has been judged scientifically suitable for publication and will be formally accepted for publication once it meets all outstanding technical requirements.

Kind regards,

Claudia Isabella Pogoreutz

Academic Editor

PLOS ONE

Additional Editor Comments (optional):

Reviewers' comments:

Reviewer's Responses to Questions

**Comments to the Author**

1. If the authors have adequately addressed your comments raised in a previous round of review and you feel that this manuscript is now acceptable for publication, you may indicate that here to bypass the “Comments to the Author” section, enter your conflict of interest statement in the “Confidential to Editor” section, and submit your "Accept" recommendation.

Reviewer #1: All comments have been addressed

Reviewer #2: All comments have been addressed

2. Is the manuscript technically sound, and do the data support the conclusions?

Reviewer #1: Yes

Reviewer #2: Yes

3. Has the statistical analysis been performed appropriately and rigorously? 

Reviewer #1: N/A

Reviewer #2: N/A

4. Have the authors made all data underlying the findings in their manuscript fully available?

Reviewer #1: Yes

Reviewer #2: Yes

5. Is the manuscript presented in an intelligible fashion and written in standard English?

Reviewer #1: Yes

Reviewer #2: Yes

6. Review Comments to the Author

Reviewer #1: All my concerns have been addressed. I believe the paper is now ready for publication.

Despite the paper not containing any new scientific method, it is an application paper that may serve well the scientific community.

Reviewer #2: he authors have responded to all raised comments, and the paper in its current format is suitable for publication.

7. PLOS authors have the option to publish the peer review history of their article (what does this mean?). If published, this will include your full peer review and any attached files.

Reviewer #1: No

Reviewer #2: No

---

## [Editor Report · Acceptance letter]

9 May 2024

PONE-D-23-14942R2 

PLOS ONE

Dear Dr. Franz, 

I'm pleased to inform you that your manuscript has been deemed suitable for publication in PLOS ONE. Congratulations! Your manuscript is now being handed over to our production team.

Kind regards, 

on behalf of

Prof. Claudia Isabella Pogoreutz 

Academic Editor

PLOS ONE